# Selective egg cell polyspermy bypasses the triploid block

Yanbo Mao[1], Alexander Gabel[2], Thomas Nakel[1], Prisca Viehöver[3,4], Thomas Baum[1], Dawit Girma Tekleyohans[1], Dieu Vo[1], Ivo Grosse[2], Rita Groß-Hardt[1]*

[1]Centre for Biomolecular Interactions, University of Bremen, Bremen, Germany; [2]Institute of Computer Science, Martin Luther University Halle-Wittenberg, Halle, Germany; [3]Faculty of Biology, Bielefeld University, Bielefeld, Germany; [4]Center for Biotechnology, Bielefeld University, Bielefeld, Germany

**Abstract** Polyploidization, the increase in genome copies, is considered a major driving force for speciation. We have recently provided the first direct in planta evidence for polyspermy induced polyploidization. Capitalizing on a novel *sco1*-based polyspermy assay, we here show that polyspermy can selectively polyploidize the egg cell, while rendering the genome size of the ploidy-sensitive central cell unaffected. This unprecedented result indicates that polyspermy can bypass the triploid block, which is an established postzygotic polyploidization barrier. In fact, we here show that most polyspermy-derived seeds are insensitive to the triploid block suppressor *admetos*. The robustness of polyspermy-derived plants is evidenced by the first transcript profiling of triparental plants and our observation that these idiosyncratic organisms segregate tetraploid offspring within a single generation. Polyspermy-derived triparental plants are thus comparable to triploids recovered from interploidy crosses. Our results expand current polyploidization concepts and have important implications for plant breeding.

*For correspondence:
gross-hardt@uni-bremen.de

## Introduction

The evolutionary history of flowering plants is characterized by recurrent polyploidization events (*Comai, 2005*; *De Bodt et al., 2005*; *Otto and Whitton, 2000*; *Van de Peer et al., 2017*). Polyploids are generally assumed to arise from unreduced gametes or somatic doubling, i.e. from defects during meiosis or mitosis (*Kreiner et al., 2017*; *Mason and Pires, 2015*; *Ramsey and Schemske, 1998*; *Sattler et al., 2016*; *Spoelhof et al., 2017*; *Tayalé and Parisod, 2013*). In addition, recent in planta and in vitro assays have provided the first direct evidence that viable polyploid plants can arise from polyspermy, the fusion of one egg cell with supernumerary sperm (*Nakel et al., 2017*; *Toda et al., 2016*). In fact, this previous work indicates that a single *Arabidopsis* plant can generate several polyspermy-induced triploid seedlings (*Nakel et al., 2017*).

The currently favored polyploidization scenario involves the formation of unreduced male gametes and the natural occurrence of such sperm has been reported for several species (*Kreiner et al., 2017*; *Mason and Pires, 2015*; *Ramsey, 2007*). Consequently, triploid plants are assumed to function as an important bridge towards polyploidization (*Comai, 2005*; *Felber and Bever, 1997*; *Ramsey and Schemske, 1998*) and field studies have identified both auto and allopolyploid triploids (*Kyrkjeeide et al., 2019*; *Lee et al., 2001*; *Marques et al., 2018*; *Meng et al., 2018*; *Schinkel et al., 2017*; *Sree Rangasamy, 1972*). However, the generation of triploid plants via unreduced male gametes is limited by the triploid block, which is a postzygotic hybridization barrier operating in many plants species (*Dilkes et al., 2008*; *Köhler et al., 2010*; *Marks, 1966*; *Ramsey and Schemske, 1998*; *Scott et al., 1998*). The triploid block is explained by the unique reproductive mode of flowering plants, which involves fertilization of two female gametes, the egg

**eLife digest** Ever since Darwin published his most famous book on the theory of evolution, scientists have sought to identify the mechanisms that drive the formation of new species. This is especially true for plant biologists who have long been fascinated by the extraordinary diversity of flowering plants.

Many species of flowering plant first evolved after a dramatic increase in the DNA content of an individual plant, a process termed polyploidization. Most explanations for polyploidization involve a pollen grain making sperm that mistakenly contain two sets of chromosomes rather than one. Yet, it is difficult to reconcile this explanation with an important aspect of plant reproduction – the so-called "triploid block".

Fertilization in flowering plants is more complicated than in animals. While one sperm fertilizes the egg cell to make the plant embryo, a second sperm from the same pollen grain must fertilize another cell to form the endosperm, the tissue that will nourish the embryo as it develops. This means that sperm with twice the normal number of chromosomes would affect the DNA content of both the embryo and the endosperm. Yet, an endosperm that receives extra paternal DNA typically halts the development of the seed via a process known as the triploid block, meaning it was not clear how often this process would actually result in a polyploid plant.

In 2017, researchers reported that plants can, on rare occasions, generate polyploid offspring via a different route: the fertilization of one egg with two sperm rather than one. Now, Mao et al. – who include several researchers involved in the 2017 study – show that this process, termed "polyspermy", can introduce extra copies of DNA into just the egg cell, meaning it can bypass the triploid block of the endosperm.

The experiments involved a model plant called Arabidopsis, and a screen of over 55,000 seeds identified about a dozen with embryos that had three parents, one mother and two fathers. Notably, most of these three-parent embryos developed in seeds that contained endosperm with the regular number of chromosomes and hence escaped the triploid block.

These new results show that polyspermy provides plants with a means to essentially sneak extra copies of DNA 'behind the back' of the DNA-sensitive endosperm and into the next generation. They also give new insight in how polyploidization may have shaped the evolution of flowering plants and have important implications for agriculture where the breeding of new "hybrid" crops has often been limited by incompatibilities in the endosperm.

and the central cell. The required sperm cell pair is typically delivered by a single pollen tube. While the fertilized egg cell gives rise to the embryo, the fertilized central cell develops into embryo-nourishing endosperm (*Russell, 1992*). Fertilization involving unreduced sperm consequently not only affects the ploidy status of the egg cell but also introduces additional paternal chromosome copies to the endosperm, and it is this latter tissue, which commonly mounts the triploid block that is manifested by seed abortion (*Köhler et al., 2010*).

In *Arabidopsis thaliana*, the effect of the triploid block is accession-dependent, being highly penetrant e.g. in Col-0, but less strict in Ler and C24 (*Dilkes et al., 2008*; *Scott et al., 1998*). A complete triploid block has been reported in many taxa (*Ramsey and Schemske, 1998*; *Schinkel et al., 2017*; *Sekine et al., 2013*; *Stoute et al., 2012*). In light of this fatal consequence, it has been suggested that there are ways to overcome this hybridization barrier (*Köhler et al., 2010*).

Making use of a two-component in planta assay, we here show that polyspermy can selectively polyploidize the egg cell, while rendering the genome size of the ploidy-sensitive endosperm unaffected. By introducing the triploid block suppressor *admetos*, we in addition show, that this unprecedented reproductive mode bypasses the triploid block.

## Results

### Establishment of a triparental embryo detection assay

Consistent with animal nomenclature, the term polyspermy is used alone when referring to egg cell polyspermy. Central cell polyspermy is specified as such. During flowering plant fertilization, both

egg and central cell fuse in a coordinated manner with a single sperm each (*Hamamura et al.,* *2011*; *Kawashima and Berger, 2011*). In order to address whether during polyspermy egg cell fertilization is still coupled to the fertilization of the central cell, we aimed at analyzing endosperm in developing seeds that contain polyspermy-derived embryos. To ease the screening process, we established a novel polyspermy-detection assay termed HIPOD$_{SCO1}$, which can efficiently and unambiguously detect the rare event of egg cell polyspermy already in developing seeds. HIPOD$_{SCO1}$ capitalizes on the pale green appearance of developing seeds defective for the gene *SNOWY* *COTYLEDON 1* (*SCO1*) (*Ruppel and Hangarter, 2007*) (*Figure 1—figure supplement 1*) and a bipartite *SCO1* complementation system, provided by two different pollen donors (*Figure 1A*). Pollen donor one contains the synthetic *GAL4* transcription factor under the control of the *RPS5a* promoter. Pollen donor two contains a functional copy of *tdTOMATO* tagged S*CO1* under the control of the GAL4 responsive *UAS* enhancer sequence. Seeds that contain a monospermy-derived embryo inherit an incomplete complementation system and will consequently be rendered pale green due to the lack of functional *SCO1*. By contrast, combinations of both constructs, which can only result from polyspermy, will give rise to green seeds with a positive tdTOMATO fluorescence signal (*Figure 1B*). It should be noted that this assay only detects polyspermy if the two sperm are derived from different pollen donors. Monopaternal polyspermy, which only delivers a single HIPOD$_{SCO1}$ component, does not rescue seed color. This scenario is expected to account for 50% of all polyspermy events, and escapes detection.

To test the system, we compared seed color of *sco1* mutant plants expressing either one of the constructs with seeds containing both, the *GAL4* activator and the *UAS* reporter line. This experiment confirmed that only the presence of both constructs complemented the defect resulting in dark green seeds, which exhibited a tdTOMATO signal (*Figure 1C*).

## Polyspermy can selectively polyploidize the egg cell

The novel HIPOD$_{SCO1}$ assay enabled us to screen for seeds that contain polyspermy-derived embryos at an advanced seed developmental stage. We processed a total of 56,493 seeds seven days after pollination (DAP) and identified 10 normally developed seeds with a change in color (*Figure 2A*, *Figure 2—figure supplement 1A*). To determine whether the candidate embryos were indeed of triparental origin, we microscopically inspected the developing seeds and found that all 10 embryos exhibited a tdTOMATO signal (*Figure 2B*, *Figure 2—figure supplement 1A*). This implies that the embryo inherited two rather than one paternal copy. To identify a corresponding shift in embryo ploidy, we carried out a chromosome spread assay. Chromosome counts are technically challenging when performed on subfractions of individual seeds and some chromosomes escape detection. However, comprehensive controls and the fact that parental chromosome contributions are quantal in nature make the assay robust and reliable.

Notably all embryos showed a triploid profile (*Figure 2C*, *Figure 2—figure supplement 1B*). This finding is comparable to the results obtained from triploid embryos segregated from an interploidy cross between diploid and tetraploid plants and contrasts with the diploid profile detected in embryos recovered from a regular cross involving haploid gametes (*Figure 2C*). To substantiate this result we introduced a GFP-tagged centromere-localized CENH3 reporter into PD1 (*De Storme* *et al., 2016*). In this complementary experiment we screened 10,774 seeds by HIPOD$_{SCO1}$ and recovered three green seeds containing tdTOMATO positive embryos (*Figure 2—figure supplement 1A*). In all seeds we detected between 11 and 15 GFP foci indicative of triploid embryos (*Figure 2E*; *Figure 2—source data 1*). Together, the analysis confirms the triparental origin of embryos in seeds with dark green color, establishing HIPOD$_{SCO1}$ as a powerful novel tool to identify polyspermy-derived embryos already in developing seeds.

In order to determine whether egg cell polyspermy is concomitant with central cell polyspermy, we assessed the ploidy of the endosperm in developing seeds containing polyspermy-induced triparental embryos. The central cell of many flowering plants, including *Arabidopsis thaliana,* is homodiploid and generates a triploid nurturing tissue after sperm fusion. In fact, we detected between 12 and 15 chromosomes in the endosperm of seeds recovered from a cross involving diploid plants. By contrast, more than 15 chromosomes are detected in control interploidy crosses between diploid female and tetraploid male (*Figure 2D*). Remarkably, in the 10 developing seeds containing triparental embryos we detected between 11 and 15 chromosomes, which is characteristic of a triploid endosperm (*Figure 2D*, *Figure 2—figure supplement 1C*). The result was substantiated by a

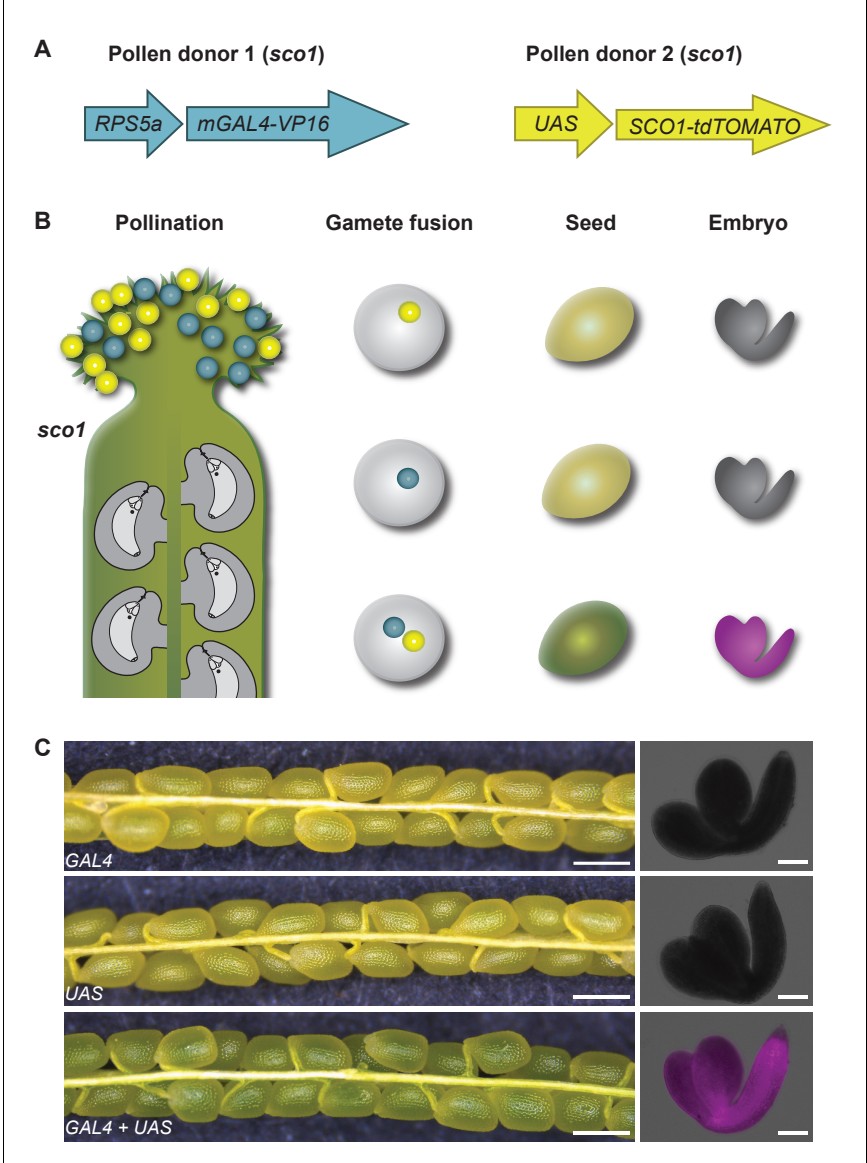

**Figure 1.** Establishment of a detection assay for polyspermy-derived embryos. (**A**) Illustration of HIPOD$_{SCO1}$. The assay is based on the UAS-GAL4 two-component system whereby a synthetic transcription factor *mGAL4* expressed under the control of the ubiquitous *RPS5a* promoter activates the *tdTOMATO*-tagged *SCO1* gene. These two components were combined with the *sco1* mutant to generate pollen donor 1 and 2 (PD1 and PD2), respectively. (**B**) Pollen of PD1 and PD2 (blue, yellow) are applied to the stigma of a *sco1* gynoecium (green). Gamete fusion involving two sperm from two different pollen donors leads to transactivation of the *SCO1* gene resulting in dark green seeds and fluorescence of tdTOMATO in the embryo, while monospermy-derived seeds remain pale green with no fluorescence. (**C**) Silique and seed analysis of *sco1* mutants containing only *pRPS5a:: mGAL4-VP16*, (upper panel), only *pUAS::SCO1-tdTOMATO* (middle panel), and both *pRPS5a::mGAL4-VP16* and *pUAS::SCO1-tdTOMATO* (lower panel). Scale bars, 500 µm and 100 µm in left and right panel, respectively. The online version of this article includes the following figure supplement(s) for figure 1:

**Figure supplement 1.** *sco1* exhibited yellowish seeds and cotyledons compared to wild-type.

complementary experiment involving the recombinant CENH3-GFP reporter, which detected a triploid profile in the endosperm of three analyzed seeds (*Figure 2F*; *Figure 2—source data 1*). Notably, at this advanced seed developmental stage, we recovered one abnormal seed from the HIPOD$_{SCO1}$ assay containing an underdeveloped triploid heart-stage embryo and tetraploid

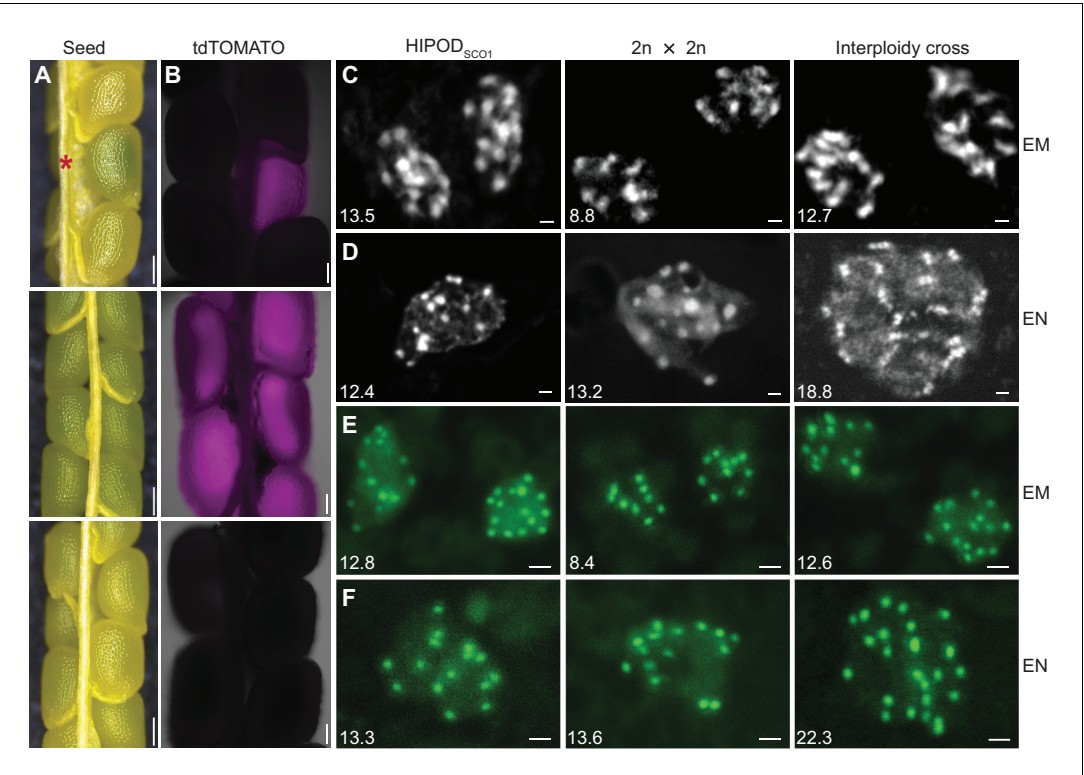

**Figure 2.** HIPOD$_{SCO1}$ identifies developing seeds harboring polyspermy-derived embryos. (**A and B**) Bright- and fluorescence light images of seeds from different crosses seven days after pollination (DAP). Upper panel (HIPOD$_{SCO1}$): *sco1 × pRPS5a::mGAL4-VP16/+ sco1* (PD1) × *pUAS::SCO1-tdTOMATO/+ sco1* (PD2); middle panel: *pUAS::SCO1-tdTOMATO/+ sco1 × pRPS5a::mGAL4-VP16/+ sco1*; lower panel: *sco1 × sco1*. Asterisk indicates polyspermy-induced complementation of a *sco1* seed. (**C and D**) DAPI-stained chromosome spreads of embryo (EM) (**C**) and endosperm (EN) (**D**) resulting from different crosses. Left panel: HIPOD$_{SCO1}$-rescued embryo segregating from cross between *sco1 × pRPS5a::mGAL4-VP16/+ sco1* (PD1) × *pUAS::SCO1-tdTOMATO/+ sco1* (PD2); middle panel: *pUAS::SCO1-tdTOMATO/+ sco1 × pRPS5a::mGAL4-VP16/+ sco1*; right panel: *sco1 × wild type* (4n). (**E and F**) Chromosome counting through centromere-targeted *CENH3-GFP* of embryo (**E**) and endosperm (**F**) resulting from different crosses. Left panel: rescued embryo segregating from cross between *sco1 × pRPS5a::mGAL4-VP16/+ p35S::CENH3-GFP sco1* (PD1 with CENH3-GFP) × *pUAS::SCO1-tdTOMATO/+ sco1* (PD2); middle panel: *pUAS::SCO1-tdTOMATO/+ sco1 × pRPS5a::mGAL4-VP16/+ p35S::CENH3-GFP sco1* ; right panel: *wild type* (4n) × *pRPS5a::mGAL4-VP16/+ p35S::CENH3-GFP sco1* . The numbers in parenthesis indicate the average counted chromosomes from all analyzed cells, from left to right, (**C**) n = 11, 11, 9, (**D**) n = 20, 6, 9, (**E**), n = 82, 57, 78, (**F**), n = 12, 26, 6. Scale bars, 200 μm (**A**), 100 μm (**B**), 1 μm (**C–E**).
The online version of this article includes the following source data and figure supplement(s) for figure 2:

**Source data 1.** Chromosome counting through centromere-targeted CENH3-GFP in embryo and endosperm resulting from different crosses.

**Figure supplement 1.** Identification and characterization of rescued green seeds and one underdeveloped seed recovered from HIPOD$_{SCO1}$.

endosperm (*Figure 2—figure supplement 1D*), characteristic of triploid block-induced seed abortion (*Dilkes et al., 2008*; *Kradolfer et al., 2013*; *Scott et al., 1998*).

Together our data indicate that egg cell polyspermy can occur independent of central cell polyspermy. Such selective polyploidization of the egg cell implies that polyspermy has the potential to bypass the triploid block.

## Most polyspermy-induced polyploidization events are insensitive to the triploid block suppressor *admetos*

To further substantiate our findings, we established a functional assay to determine the potential of polyspermy in bypassing this reproductive barrier. It was previously shown that mutations in the

paternally expressed imprinted gene *ADMETOS (ADM)* suppress the triploid block in *Arabidopsis thaliana* (*Kradolfer et al., 2013*). In fact, interploidy crosses between diploid and tetraploid plants lead to a 15.6 fold increase in fertile triploid seeds when the tetraploid pollen donor segregated the *adm-1* allele (*Figure 3A–C*, *Figure 3—source data 1*). If polyspermy would equally trigger the triploid block, we would expect a similar increase in polyspermy frequencies when using *adm-1* segregating pollen donors. In order to identify polyspermy-derived seedlings, we made use of the previously established HIPOD assay that works analogous to the HIPOD$_{SCO1}$ system introduced above, but positively selects triparental seedlings on the basis of herbicide resistance (*Nakel et al., 2017*). A total of 116,279 and 113,777 seeds were harvested from three independent HIPOD experiments using either *adm-1* or wild-type segregating pollen donors, respectively. Out of these, 47 herbicide resistant seedlings segregated from pollen donor with *adm-1* background while 27 were recovered from wild type (*Figure 3D–F*, *Figure 3—source data 1* ). This corresponds to an almost two fold increase in the *adm-1* segregating approach (*Figure 3C*), which is more than eight times lower than the effect observed in the interploidy cross.

Previous results suggested that the egg cell block is stricter than the central cell block (*Grossniklaus, 2017*; *Scott et al., 2008*) and fertilization of the two female gametes during monospermy has been shown to occur in a coordinated fashion (*Kawashima and Berger, 2011*; *Hamamura et al., 2011*). Our unprecedented finding that most polyspermy-derived embryos develop in the presence of a monospermy-derived endosperm show that polyspermy enables selective polyploidization of the egg cell and concomitant bypassing the triploid block.

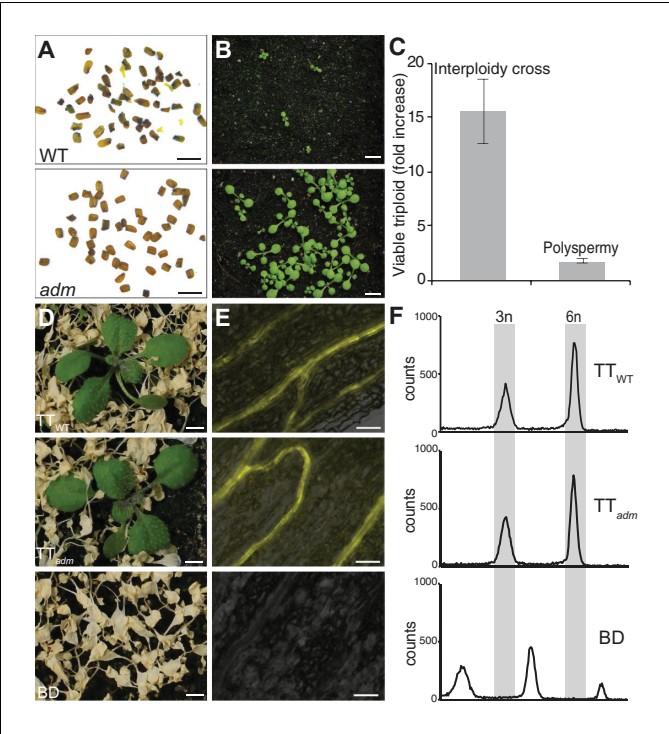

**Figure 3.** Polyspermy-induced polyploidization is partially insensitive of *adm*-mediated triploid block repression. (**A and B**) Mature seed (**A**) and corresponding 9 day old seedling (**B**) from one silique of wild-type pollinated with diploid wild-type pollen and diploid *adm-1* pollen. (**C**) Effect of the triploid block repressor *adm* on interploidy cross- recovered triploids (2n × 4n) and polyspermy- derived triploids. Shown is the ratio of triploid plants recovered from two crosses involving either *adm* or wild-type pollen donors (*adm*/WT). The data are means ± SEM (n = 3 experiments). (**D**) Herbicide-treated offspring of triparental triploid (TT) plants recovered from HIPOD with (TT$_{WT}$) or without (TT$_{adm}$) *ADMETOS* segregating pollen donors. Lower panel, herbicide-sensitive offspring of biparental diploid wild type (BD). (**E and F**) YFP fluorescence (**E**) and flow-cytometric analysis (**F**) of TT$_{WT}$, TT$_{adm}$ and BD plants corresponding to the categories shown in (**D**). Scale bars, 1 mm (**A**), 5 mm (**B, D**), 50 μm (**E**).
The online version of this article includes the following source data for figure 3:

**Source data 1.** Comparison of viable triploids recovered from polyspermy and interploidy crosses (2n × 4n).

## Polyspermy-derived triparental plants are comparable to triploids generated by interploidy crosses

The transcript profile of triploid plants has been characterized previously and remarkably few differences were found with respect to their cognate diploid controls (*Hou et al., 2018*). Polyspermy-induced plants differ from triploids derived from interploidy crosses as they inherit two sperm cytoplasms and, as shown in this work, mostly develop in a seed characterized by identical ploidies in embryo and endosperm. Given their special mode of origin, we aimed to compare the transcript profile of polyspermy-induced triparental triploids (TT) with that of biparental triploids (BT). In addition, we compared the transcriptome profile of BT plants with that of biparental diploids (BD) to identify ploidy-dependent changes in the transcriptional landscape. We used *ein3* mutants as pollen acceptor as they have previously been shown to attract supernumerary pollen tubes (*Völz et al., 2013*). We performed RNAseq on five plants 18 days after sowing (DAS) each from BD, BT, and TT (*Figure 4—figure supplement 1A*). We chose this early state as we expected potential differences to become established during seed development, which differs in the three settings.

In order to assess the quality of the transcriptome data, and to identify potential transcriptome-wide differences between the mRNA profiles of the three groups of plants, we performed a hierarchical clustering, a two-dimensional principal component analysis (PCA), and a two-dimensional multidimensional scaling (MDS) analysis of the fifteen 21,450-dimensional normalized expression profiles. We found that Spearman's correlation coefficient $c$ was greater than 0.94 for all of the 105 pairs of profiles (*Figure 4—figure supplement 1B*). Notably, the sub-trees of the dendrogram obtained by hierarchical average linkage clustering did not correspond to the five biological replicates from the same genotype (*Figure 4A*), and the 15 samples showed a high overlap in the PCA (*Figure 4B*) and MDS (*Figure 4—figure supplement 1C*) plots. These results not only reflect the high quality of the transcriptome data but also suggest a high similarity of the 15 transcriptome profiles compared.

We first addressed, if there were genes with ploidy-specific expression changes by comparing transcript profiles of BD and BT. This comparison did not yield a single gene with a statistically significant differential expression. This result is in support of previous transcriptome profiling approaches that have uncovered remarkably few changes in plants with different ploidy (*Pignatta et al., 2010*; *Riddle et al., 2010*; *Stupar et al., 2007*; *Wang et al., 2006*; *Yu et al., 2010*).

We next compared the transcript profiles of BT and TT in order to identify specific expression changes potentially associated with polyspermy. Interestingly, also this approach did not yield genes with statistically significant differential expression. The similarity in the overall transcriptional profiles between TT and BT plants is reflected by strong similarities in various life-history traits, including flower organ size, cell size, and even fertility (*Figure 4—figure supplement 2*). Please note that the normalized expression data presented here does not allow any conclusions on alterations in transcriptome size, i.e. changes affecting the total number of transcripts per cell (*Coate and Doyle, 2015*). However, the data suggests that Arabidopsis responds in a transcriptionally balanced fashion to the inheritance of supernumerary genomes and seed homoploidy.

Along these lines, also the ability to generate tetraploid offspring within a single generation, a parameter that has been described previously for interploidy cross-induced triploids was maintained: To assess whether polyspermy-derived triploids can segregate stable polyploid offspring, we harvested the seeds of polyspermy-derived triparental plants and propagated them in two successive generations. The progeny of polyspermy-derived triploids segregates a complex swarm of karyotypes, similar to what has previously been described for interploidy crosses (*Henry et al., 2005*). On the basis of flow cytometric analysis, we grouped the plants into five different categories: near-diploids, 2n-3n, near-triploids, 3n-4n and near-tetraploids (*Figure 4C*, *Figure 4—figure supplement 3A*). Already in the $F_2$ generation 5 out of 109 plants were found to fall into the near-tetraploids category, while 22 plants segregated a diploid-like profile (*Figure 4C*, *Figure 4—figure supplement 3B*). To determine whether any of the high-ploidy plants represented a genuine tetraploid, we collected seeds and determined the ploidy of 20 offspring per individual $F_2$ plant. Flow cytometric analysis revealed that 3 out of 5 near-tetraploid $F_2$ plants segregated exclusively plants that exhibit a ploidy profile characteristic to 4n plants (*Figure 4D*). This result was confirmed in the $F_4$ generation, which, again, revealed a homogenous tetraploid ploidy profile. Together these results show that

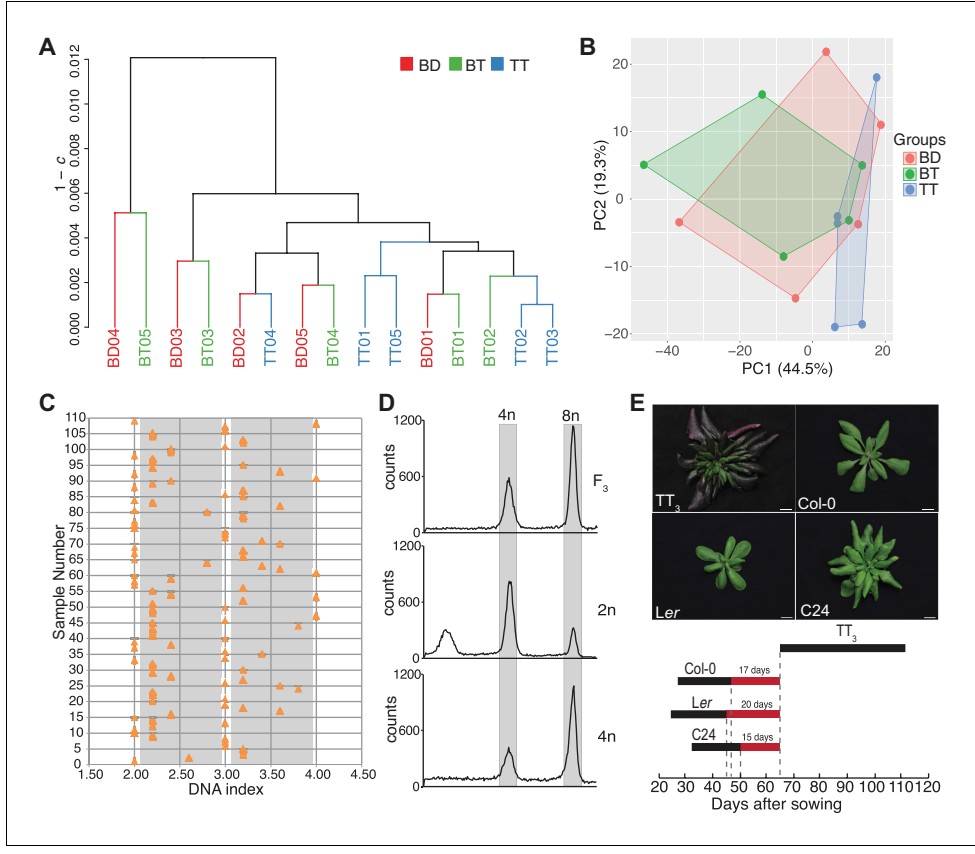

**Figure 4.** Characterization of triparental triploid plants. (**A**) Dendrogram from hierarchical clustering of the 3 × 5 biological replicates of biparental diploid (BD), biparental triploid (BT), and triparental triploid (TT) recovered from polyspermy plant samples. The 15 × 15 matrix of Spearman's correlation coefficients $c$ was computed from the fifteen regularized log transformed expression profiles. The dendrogram was computed by average linkage clustering (UPGMA) of the 15 × 15 dissimilarity matrix with elements $1 − c$. (**B**) Two-dimensional principle component analysis (PCA) of the fifteen 21,450-dimensional regularized log transformed expression profiles. The points represent the biological replicates, while the shapes resulting from connecting biological replicates from the same genotype highlight the similarity between the transcriptome profiles. (**C**) Distribution of ploidy level in the aneuploidy swarms produced by a triparental triploid plant. 2.0 in DNA index represents near diploids, 3.0, near triploids, and 4.0, near tetraploids. The gray areas indicate the intermediate ploidies. Each orange triangle represents an $F_2$ plant derived from a triparental triploid. (**D**) Flow cytometric analysis of tetraploid progeny plants in $F_3$ generation. 2n and 4n represent diploid and tetraploid controls. (**E**) Analysis of flowering time window of triparental triploid plants recovered from a three accession cross ($TT_3$), Col-0, L*er* and C24 during bolting stage. Scale bar, 1 cm. The black bold line represents the flowering period of different accessions. The three red bold lines bordered by the gray dashed lines label the day gaps between flowering termination of the parents and flowering initiation in the $TT_3$.

The online version of this article includes the following figure supplement(s) for figure 4:

**Figure supplement 1.** Transcriptome analysis of plants from different origin and ploidy.
**Figure supplement 2.** Polyspermy-induced triploid plants are similar to triploid plants originating from a regular monospermic fertilization mode.
**Figure supplement 3.** Ploidy assessment of offspring derived from triparental triploids.
**Figure supplement 4.** Flowering initiation and termination comparison.

polyspermy-derived triploid plants have the potential to generate stable tetraploid and diploid offspring within a single generation.

## Polyspermy-derived three accession hybrids are reproductively isolated from their parental lines

We previously combined three distinct Arabidopsis accessions in a three parent cross (*Nakel et al., 2017*) and noticed that the resulting triploid hybrids initiate flowering later than their parents, an effect which was previously described also for two-accession hybrids (*Groszmann et al., 2014*; *Moore and Lukens, 2011*) (*Figure 4—figure supplement 4A*). To address whether this phenotype has the potential to reduce gene flow, we compared flowering time between the parental line and triparental three accession hybrids, henceforth referred to as $TT_3$. We found that flowering is induced more than one month later in $TT_3$ than in the parental lines, and the flowering period of $TT_3$ is completely isolated from the parents (*Figure 4E*). Under our plant growth conditions, Ler and Col-0 start to flower $24.4 \pm 0.7$ and $26.6 \pm 1.0$ DAS, and flowering terminates $44.9 \pm 1.2$ and $47.1 \pm 0.9$ DAS, respectively. This corresponds to a flowering period of around 20 days. The C24 flowering window is comparable but flower initiation is delayed by five days ($31.8 \pm 1.2$ DAS). By contrast, $TT_3$ plants initiate flowering only after $64.6 \pm 5.4$ days, which is around 20, 17 and 15 days after the respective parental lines have terminated their flowering phase (*Figure 4E*; *Figure 4—figure supplement 4B*). Even though these data are obtained under optimized growth condition, the results suggest that polyspermy-derived triploid three accession hybrids are reproductively isolated from the parental plants in the first generation.

## Discussion

The triploid block is an established and widely distributed postzygotic hybridization barrier. In light of its fatal consequence, it has been suggested that there are ways to overcome this hybridization barrier (*Köhler et al., 2010*). We here established a novel polyspermy detection assay that allows to identify and characterize developing embryos resulting from supernumerary sperm fusion. With this tool we were able to show that most polyspermy-derived plants develop from seeds resulting from selective egg cell polyploidization. In those seeds, supernumerary paternal copies are only transmitted to the embryo, thereby bypassing the triploid block of the endosperm. Our results expand previous polyploidization concepts, which state that the increase in genome copies is caused by infrequent meiotic or mitotic defects. In fact, the currently favored route towards polyploid plants involves unreduced male gametes; however, this scenario introduces supernumerary paternal copies also to the endosperm, which is not tolerated in many plants resulting in seed abortion (*Dilkes et al., 2008*; *Ramsey and Schemske, 1998*; *Scott et al., 1998*; *Stoute et al., 2012*). In fact, this endosperm-related triploid block is considered a means of reproductive isolation (*Köhler et al., 2010*; *Ramsey and Schemske, 1998*). Plant polyploidization via polyspermy, by contrast, often affects the embryo-derived seed fraction only and hence has the potential to bypass the triploid block. It will be a challenge for the future to determine whether and to what extent polyspermy is relevant in nature and contributed to the evolution of polyploid plants.

From an evolutionary and agricultural point of view, another intriguing implication of polyspermy-induced polyploidization is the one-step combination of three genetically distinct plant genomes in a three-parent hybridization. This scenario is expected to occur rarely in nature, even when taking into account wind and animal pollination. Still, it might be relevant in an evolutionary time scale and when considering the huge seed amounts produced by many plants. In addition, our findings might have important implications for agriculture: Not only do three-parent-crosses hold the potential to instantly combine beneficial traits of three rather than two cultivars. They, in addition, might provide a means to overcome endosperm-induced hybrid incompatibilities by transmitting incompatible sperm along with compatible sperm through the egg cell only. Scrutinizing the potential of this genetic hitchhiking approach opens up new avenues for future research.

## Materials and methods

### Plant materials and growth conditions

Unless otherwise stated, all experiments were carried out using *Arabidopsis thaliana* Columbia (Col-0). The *sco1* T-DNA insertion mutant (*SALK_025112*) (*Ruppel and Hangarter, 2007*) was obtained from European Arabidopsis Stock Center (NASC) (Nottingham, UK). The *adm-1* mutant

(*Kradolfer et al., 2013*) was kindly provided by Claudia Köhler. For RNA-Seq, *ein3-1* mutant (*Völz et al., 2013*) was used as female plant. Triparental triploids (TT) were obtained from double pollination as previously described (*Nakel et al., 2017*), while biparental triploids (BT) were recovered from a cross of 2n female with 4n male pollen donor *pRPS5a::mGAL4-VP16/-; pUAS::BAR-YFP/-*. Biparental diploids (BD) were generated from a cross of 2n female with 2n pollen donor *pRPS5a::mGAL4-VP16/-; pUAS::BAR-YFP/-*. Stratified seeds (2 days, 4°C) were germinated on soil in a Conviron MTPS growth chamber under long-day conditions (16 hr light/8 hr dark) at 23°C. For HIPOD$_{WT}$, HIPOD$_{adm}$ and experiments in *Figure 4—figure supplement 2*, plants were transferred to 18°C after bolting.

## Colchicin-induced polyploidization

Tetraploid Col-0 wild-type, *adm-1/-* and pollen donor plants were generated by using 0.05% colchicine treatment following the one-drop method previously described (*Yu et al., 2009*).

## PCR-based genotyping

Primer sequences are given in *Supplementary file 1*. The *sco1/-* mutant was confirmed via PCR amplification using primer pair LBb1.3/YM8-RP and YM8-RP/YM7-LP for the mutant and wild-type allele, respectively. The *adm-1* allele was identified as previously described (*Kradolfer et al., 2013*). The presence of *pRPS5a::mGAL4-VP16* and *pUAS::BAR-YFP* was confirmed by using TN12s/TN12as and TN26s/TN26as, respectively.

## Plasmid construction and transgenic lines

The *pUAS::SCO1-tdTOMATO::tNOS* vector was obtained by inserting YM1-F/YM2-R amplified CDS of *SCO1* from cDNA library of leaves into DR13 (*pAt5g40260::NLS-tdTOMATO::tNOS*) (*Völz et al., 2013*) using PacI/NotI, followed by exchanging *pAt5g40260* with *pUAS* sequence from *pUAS::BAR-YFP* (*Nakel et al., 2017*) using AscI/PacI.

The *CENH3* CDS was amplified from leaf cDNA library using primer pair DT231s/DT231as and cloned into *p35S::NLS-GFP* using AvrII and MfeI to generate *p35S::CENH3-GFP*.

Transgenic Arabidopsis plants were generated by following the floral dip protocol using *Agrobacterium tumefaciens* (*Clough and Bent, 1998*). Pollen donors with *adm-1* background were generated by crossing *adm-1* mutant with each pollen donor.

## Germination and flowering time assay

For germination assay, all seeds from a single silique (n $\geq$ 8) were sown on soil. Germination frequency was scored after 9 days. The experiment was repeated three times.

Flowering time was assessed as days after sowing (DAS), starting from the day of sowing till the opening of the first flower. Within three independent experiments a total of 7 plants (TT$_3$), 36 (Col-0), 37 (L*er*), and 35 (C24) were analyzed. Flowering period was defined as the interval between the formation of the first and the last flower. In independent experiments a total of 4 plants (TT$_3$), 36 (Col-0), 37 (L*er*), and 35 (C24) were analyzed. Please note that watering was stopped once the main branch stopped flowering.

## HIPOD$_{SCO1}$

For the HIPOD$_{SCO1}$ experiment, 2–3 closed flower buds from one inflorescence of *sco1/-* mutant were emasculated. 3 days after emasculation, pollination was carried out using pollen grains collected individually from plants of *sco1/- pRPS5a::mGAL4-VP16/+* and *sco1/- pUAS::SCO1-tdTO-MATO/+*. These two pollen categories were applied onto the stigmatic surface with two brushes. 7 days after pollination (DAP), siliques were dissected under Leica S6E or S8apo stereomicroscope (Leica, Germany) for seed color screening. Seeds with a change in color were analyzed for tdTO-MATO signal and embryo and endosperm chromosome number.

## Ploidy analysis

Sample preparation and ploidy determination was performed using the same method as described previously; nuclei were marked with CyStain UV Precise P kit and assessed with CyFlow ploidy analyzer (*Nakel et al., 2017*). The triploid swarm ploidies were analyzed by mixing nuclei extracted

from two similar sized leaf pieces derived from a diploid control plant and a leaf sample of TT progeny.

## RNAseq

Seeds that contain TT, BT, and BD were selected on ½ MS medium containing 25 µg/ml Glufosinate-ammonium (PPT, Sigma), and the seedlings were transferred to soil 9 DAS. YFP and ploidy analysis was carried out 17 DAS to confirm TT, BT and BD profile. 18 DAS, aerial tissues were frozen and grinded in pre-cooled EP tubes with metal beads and total RNA was isolated following the procedure of GeneMATRIX Universal RNA purification kit.

750 ng total RNA per sample was used to prepare sequencing libraries according to the Illumina TruSeq RNA Sample Preparation v2 Guide. Purification of the poly-A containing mRNA was performed using two rounds of poly-T Oligos attached to magnetic beads. During the second elution of the poly-A RNA, the RNA was fragmented and primed for cDNA synthesis. After cDNA synthesis the fragments were end-repaired and A-tailing was performed. Multiple indexing adapters were ligated to the ends of the cDNA fragments and the adapter ligated fragments were enriched by 10 cycles of PCR. After quali- and quantification the resulting sequencing libraries were pooled equimolar and sequenced 75 bp single-read on an Illumina NextSeq500.

## Transcript profiling

The sequenced single-end reads were processed by *trimmomatic version 0.38* (*Bolger et al., 2014*). We used the parameters *SE -phred33 ILLUMINACLIP: adapter_sequences.fasta:2:30:10* to remove remaining adapter sequences. For performing a quality trimming, we used the parameters *SE -phred33 LEADING:5 TRAILING:5 SLIDINGWINDOW:4:15 MINLEN:36*. Next, the trimmed reads were mapped to the *Arabidopsis thaliana* genome from EnsemblGenomes (*Kersey et al., 2018*), release 36, by *STAR version 2.6.0c* (*Dobin et al., 2013*) with parameters *–alignIntronMin 20 –alignIntronMax 50000 –outFilterMismatchNmax 2 –outFilterMultimapNmax 50 –outFilterIntronMotif*s. The assignment of reads to the annotated genes of EnsemblGenomes, release 36, and the quantification of transcript expression were performed by *salmon version 0.11.3* (*Patro et al., 2017*) with parameters *quant –libType U –gcBias –seqBias –useErrorModel*.

Genes not having more than zero counts in at least three out of five biological replicates in at least one genotype were removed from the transcriptome analysis, resulting in a count table of 21,450 genes. Hence, we still keep genes in the analysis being expressed in just one genotype. The normalization of counts, the computation of the regularized log transformation, and the identification of differentially expressed genes were performed by the R package (*R Development Core Team, 2018*) DESeq2 (*Love et al., 2014*).

For each pair of samples, we computed Spearman's correlation coefficient $c$ based on the regularized log transformed count data, and we computed the dendrogram of the 15 samples by hierarchical average linkage clustering with the distances $d = 1 – c$.

Principle component analysis (PCA), multidimensional scaling analysis (MDS), the computation of Spearman's correlation coefficients, and hierarchical clustering were performed by functions *prcomp*, *cmdscale*, and *cor* of the R package *stats* as well as by the UPGMA algorithm implemented in the R function *hclust*. Alternatively to the PCA, we chose the MDS analysis as a non-linear approach for dimensional reduction that tries to preserve as closely as possible the pairwise distances of the original multidimensional data points.

The prediction of differentially expressed genes was performed by the Wald test, implemented in *DESeq2*, with a subsequent application of the Benjamini-Yekutieli correction procedure and a threshold of the false discovery rate of 5%.

## Chromosome spread

7 DAP seeds were harvested and their embryo (cotyledon stage) was extracted by applying pressure with syringe needles. The separated embryo and the remaining part of the seed containing the endosperm was individually transferred into 200 µl pre-treatment solution and incubated for 4 hr at RT (*Scott et al., 2008*). For aborted seeds recovered from interploidy crosses and HIPOD$_{SCO1}$, the embryo was separated from the endosperm in the final fixative step on the slide. The remaining steps were carried out as described in *Armstrong et al. (2001)*.

## Microscopy

Images were taken using Leica DMI6000b epifluorescence inverted microscope, equipped with YFP, DAPI and DsRed filter cube. For YFP expression analysis, dissected sepals were transferred to 10% glycerol and data were collected. For chromosome spread, data from DAPI stained samples were collected with the aid of 100 × oil objective and an immersion oil Immersol 518F (Zeiss, Germany) or by using confocal laser scanning microscopy with Airyscan module (Zeiss LSM 880) at excitation of 405 nm and emission of 421 nm. CENH3-GFP localization was observed at excitation of 488 nm and emission of 510 nm. 7 DAP silique images were captured by using Leica S8apo stereomicroscope.

## Acknowledgements

We thank Claudia Köhler for providing the *adm-1* seeds, Bernd Weisshaar for providing the genomics platform at the Center for Biotechnology (CeBiTec) at Bielefeld University. We gratefully acknowledge Claudia Köhler, Bernd Weisshaar and members of the Groß-Hardt lab for valuable comments on the manuscript. We gratefully acknowledge the financial support from the European Research Council to RG (ERC Consolidator Grant "bi-BLOCK" ID. 646644) and the German Research Foundation to IG (DFG, GR3526/8).

## Additional information

### Competing interests

Rita Groß-Hardt: The authors declare that they have filed a patent based on this work (EP3485020 A1, CN109790546, US2019159417 A1). The other authors declare that no competing interests exist.

### Funding

| Funder | Grant reference number | Author |
|---|---|---|
| H2020 European Research Council | 646644 | Rita Groß-Hardt |
| Deutsche Forschungsgemeinschaft | GR 3526/8 | Ivo Grosse |

The funders had no role in study design, data collection and interpretation, or the decision to submit the work for publication.

### Author contributions

Yanbo Mao, Thomas Nakel, Conceptualization, Data curation, Formal analysis, Validation, Investigation, Visualization, Methodology, Writing - review and editing; Alexander Gabel, Conceptualization, Data curation, Software, Formal analysis, Validation, Investigation, Visualization, Methodology, Writing - review and editing; Prisca Viehöver, Resources, Data curation, Software, Investigation, Methodology, Writing - review and editing; Thomas Baum, Data curation, Formal analysis, Validation, Investigation, Visualization, Methodology, Writing - review and editing; Dawit Girma Tekleyohans, Conceptualization, Validation, Investigation, Methodology, Writing - review and editing; Dieu Vo, Formal analysis, Investigation, Visualization; Ivo Grosse, Conceptualization, Resources, Data curation, Software, Formal analysis, Supervision, Funding acquisition, Validation, Investigation, Visualization, Methodology, Project administration, Writing - review and editing; Rita Groß-Hardt, Conceptualization, Resources, Data curation, Formal analysis, Supervision, Funding acquisition, Validation, Investigation, Visualization, Methodology, Writing - original draft, Project administration, Writing - review and editing

### Author ORCIDs

Yanbo Mao ⓘD https://orcid.org/0000-0001-5520-8202
Alexander Gabel ⓘD https://orcid.org/0000-0002-8064-3289
Thomas Nakel ⓘD https://orcid.org/0000-0001-9033-5987
Thomas Baum ⓘD http://orcid.org/0000-0002-4409-2635

Dawit Girma Tekleyohans (iD) https://orcid.org/0000-0001-7383-5971
Ivo Grosse (iD) http://orcid.org/0000-0001-5318-4825
Rita Groß-Hardt (iD) https://orcid.org/0000-0003-1998-0507

## Decision letter and Author response

Decision letter https://doi.org/10.7554/eLife.52976.sa1
Author response https://doi.org/10.7554/eLife.52976.sa2

## Additional files

### Supplementary files
- Supplementary file 1. List of primers used for PCR.
- Transparent reporting form

### Data availability

All data generated or analyzed during this study are included in this article (and the additional files). The sequencing data is available from the NCBI's Gene Expression Omnibus database under accession number GSE130186.

The following dataset was generated:

| Author(s) | Year | Dataset title | Dataset URL | Database and Identifier |
|---|---|---|---|---|
| Yanbo Mao, Alexander Gabel, Thomas Nakel, Prisca Viehöver, Thomas Baum, Dawit Girma Tekleyohans, Dieu Vo, Ivo Grosse, Rita Groß-Hardt | 2020 | Selective egg cell polyspermy bypasses the triploid block | https://www.ncbi.nlm.nih.gov/geo/query/acc.cgi?acc=GSE130186 | NCBI Gene Expression Omnibus, GSE130186 |

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
