## [Decision Letter]

**Acceptance summary:**

As mentioned in our previous decision letter, we and the reviewers considered your work as a conceptual breakthrough, providing highly novel insights and concepts into reproduction system in angiosperms. Your work highlights that polyspermy can circumvent the triploid block in the endosperm and moreover clearly illustrate that polyspermy-derived triploid plants have the potential to form tetraploid progenies in one generation.

**Decision letter after peer review:**

Thank you for submitting your article "Polyspermy selectively polyploidizes the egg cell thereby bypassing the triploid block" for consideration by *eLife*. Your article has been reviewed by three peer reviewers, and the evaluation has been overseen by Reviewing Editor Jürgen Kleine-Vehn and Christian Hardtke as the Senior Editor. The following individual involved in review of your submission has agreed to reveal their identity: Takashi Okamoto (Reviewer #2).

The reviewers have discussed the reviews with one another and the Reviewing Editor has drafted this decision to help you prepare a revised submission.

The reviewers considered your work as a conceptual breakthrough, providing highly novel insights and concepts into reproduction system in angiosperms. Your work highlights that polyspermy can circumvent the triploid block in the endosperm and moreover clearly illustrate that polyspermy-derived triploid plants have the potential to form tetraploid progenies in one generation.

The reviewers appreciated the high quality of your data and did not raise any major concerns that would require experimental validation. Please consider the reviewer comments (see below) to further strengthen the manuscript. Specifically, please clearly state/discuss the mechanistic advance of this work in relation to your previous publication (Nakel et al., 2017).

Reviewer #1:

How polyploidization comes about is a central question in biology. In plants polyploidization is frequently associated with speciation and occurred during the generation of important crop plants, such as wheat. Thus, the authors address a question of high scientific significance and of broad interest.

In previous work the authors established that polyploidization in plants can occur via the rare occurrence of polyspermy due to more than one pollen donor contributing to fertilization (as a rule only one pollen tube fertilizes a given ovule). In my opinion, the central advance provided by this paper relates to their finding that during this process two sperm cells, in their set up derived from genetically different pollen donors, fertilize the egg cell. By contrast, the central cell, the other gamete, is fertilized by only one sperm cell. Their data confirm that the resulting embryo is triploid while the endosperm, which develops from the fertilized central cell, is triploid. The authors go on to show that by this mechanism polyspermy circumvents the triploid block, a postzygotic polyploidization barrier, which occurs in the endosperm. It is noteworthy that their genetic method is based on an unbiased approach and does not select for the event, for example through screening for herbicide resistance.

Overall, I find this a very exciting paper. It certainly has ramifications as to how polyploidization can come about. Moreover, it has obvious potential implications for evolutionary biology (and application), although I am not sure how relevant that really is given the low observed frequency of the event. I concede, however, that I am not an expert in evolution and speciation.

Reviewer #2:

Polyploidization has played a major role in the long-term diversification and evolutionary success of angiosperms. Triploid formation among diploid plants, which is generally considered to be achieved by fertilization of an unreduced gamete with a reduced one, has been accepted as a means of polyploid production. However, fusion of an unreduced sperm cell with a reduced central cell forms primary endosperm cell with unbalanced parental genome ratio (maternal genome: paternal genome = 2m: 2p), and such primary endosperm cell abnormally develops, resulting in arrest of embryogenesis and seed development.

Recently, emergence of polyspermy-derived triploid embryo (progeny) has been clearly presented in planta (Nakel et al., 2017) and in vitro (Toda et al., 2016). Moreover, using a novel GAL4/UAS system, termed HIPOD system, it was clearly indicated that triparental progeny is produced through hetero-pollination (polyspermy) in Arabidopsis by Dr. Groß-Hardt's group (Nakel et al., 2017). The present study is continuous work of the same group, and the authors successfully expanded the HIPOD system to HIPOD-SACO system to analyze developmental profiles of polyspermy-derived embryos and seeds.

In the study, authors showed that hetero-pollination in Arabidopsis resulted in production of seed in which polyspermy-derived triploid embryo (1m:2p) and monospermy-derived endosperm (2m:1p) are formed. This finding is extremely important and interesting, since triploid progeny (embryo) can be developed without effect of triploid block from unbalanced endosperm. In addition, it was also indicated that tetraploid seeds were obtained from self-pollination of the polyspermy-derived triploid plants. This suggests that polyspermy-derived triploid plants has potential to form tetraploid progenies in one generation, being consistent with the triploid-bridge pathway for emergence of tetraploid plants (polyploid plants). Therefore, the contents of the present study provide highly novel insights and concepts into reproduction system in angiosperms, and the reviewer considers that the present study will be a new milestone for basic and applied plant biology. Therefore, the manuscript will be suitable for publication in *eLife*. Comments from the reviewer are presented below, and the authors can address the comments.

Does polyspermy selectively polyploidize egg cell? (The term "selective" in title, Abstract and at the end of the subsection “Polyspermy-induced polyploidization events are often insensitive to triploid block suppression by *admetos*”).

The reviewer totally agrees that triploid-block can be escaped in the seeds containing polyspermy-derived 1m:2p embryo and monospermy-derived 2m:1p endosperm, and such seeds with triploid progeny develop normally. However, judging from the results of HIPOD experiments using *adm1* and wild-type pollen donors (subsection “Polyspermy-induced polyploidization events are often insensitive to triploid block suppression by *admetos*”), it can be considered that the polyspermy occurs not only in egg cells, but also in central cells with nearly equal efficiency. Two-fold numbers of possible triploid progenies (seeds) were obtained when *adm1* donors were employed, compared with wild type donors. This may indicate the possibility that polyspermy occurs both in egg cell and central cell evenly also in case of using wild-type donors, and that, among these variable kinds of fertilized ovaries (seeds), only half of triploid seeds containing 1m:2p embryo and 2m:1p endosperm can survive to mature seeds but remaining half seeds possessing 2m:2p endosperms aborts. In this context, the reviewer feels that the terms 'selective polyploidization' will not be appropriate for presenting the polyspermy-situation in the study. 'Polyspermy-derived selective growth of triparental progeny via triploid block escape' may correctly sound.

Reviewer #3:

This study investigates a potential cause of whole genome duplication and polyploidy in plants as polyspermy (fertilization of an egg by more than one sperm cell). Polyploids in plants are known in many cases to arise from production of unreduced gametes, caused by a change in the progression of meiosis. However, fertilization by unreduced male gametes typically leads to triploid block caused by genome inbalance in the endosperm.

The authors develop a specialized assay HIPOD-sco1 to detect polyspermy. This uses a mutation called sco1 that causes seed to be pale green. This is complemented by two different transgenes; (i) RPS5a::GAL4 and (ii) UAS::Tomato-SCO1. In the case of polyspermy both transgenes may be inherited causing complementation of sco1 seed phenotype and expression of the fluorescent TOM-SCO1 protein. I assume pollen is mixed from the two donors and then applied to the sco1 mutant flowers?

The authors screened 56K seeds and found 10 seeds that showed evidence of complementation and tomato expression. Using spreads and labeling with CENH3-GFP the authors are able to confirm that the rescued embryos are triploid, and the endosperm is of expected triploid genotype (as in wild type), which implies that polyspermy of the egg cell and not the central cell occurred.

Mutations in the ADM gene suppress the triploid block in Arabidopsis. The authors report – the results of this experiment (subsection “Polyspermy-induced polyploidization events are often insensitive to triploid block suppression by *admetos*”) – is the data seen in *adm1* significantly different from wild type. The comparison with data from interploidy crosses is also hard to interpret from the way this is presented.

Next the authors perform transcript profiling in triploid plants and consistent with published work observe few if any significant changes.

Although the work is well performed – my major reservation is that I don't think a sufficient advance is made over previous work (Nakel et al., 2017) by the same authors.

---

## [Author Response]

Reviewer #2:[…] Does polyspermy selectively polyploidize egg cell? (The term "selective" in title, Abstract and at the end of the subsection “Polyspermy-induced polyploidization events are often insensitive to triploid block suppression by admetos”).The reviewer totally agrees that triploid-block can be escaped in the seeds containing polyspermy-derived 1m:2p embryo and monospermy-derived 2m:1p endosperm, and such seeds with triploid progeny develop normally. However, judging from the results of HIPOD experiments using adm1 and wild-type pollen donors (subsection “Polyspermy-induced polyploidization events are often insensitive to triploid block suppression by admetos”), it can be considered that the polyspermy occurs not only in egg cells, but also in central cells with nearly equal efficiency. Two-fold numbers of possible triploid progenies (seeds) were obtained when adm1 donors were employed, compared with wild type donors. This may indicate the possibility that polyspermy occurs both in egg cell and central cell evenly also in case of using wild-type donors, and that, among these variable kinds of fertilized ovaries (seeds), only half of triploid seeds containing 1m:2p embryo and 2m:1p endosperm can survive to mature seeds but remaining half seeds possessing 2m:2p endosperms aborts. In this context, the reviewer feels that the terms 'selective polyploidization' will not be appropriate for presenting the polyspermy-situation in the study. 'Polyspermy-derived selective growth of triparental progeny via triploid block escape' may correctly sound.

Thank you for this comment. It is important to discriminate between the seeds generated by selective egg cell polyspermy/polyploidization and the polyploid plants recovered. The latter category commonly traces back to egg cell polyspermy only, because the category of egg and central cell polyspermy, is negatively selected during seed development by the triploid block. However, we understand the concern of this reviewer and agree that this can be misleading. We have accordingly adjusted title and text:

Title: "Selective egg cell polyspermy bypasses the triploid block".

Abstract: We introduced "can" and modified the sentence describing the *adm* result: "In fact, we here show that most polyspermy-derived seeds are insensitive to the triploid block suppressor *admetos*." In the remaining text we introduced "can" in the last paragraph of the Introduction and in the subheading “Polyspermy can selectively polyploidizes the egg cell”, and "most" in the last sentence of the subsection “Most Polyspermy-induced polyploidization events are often insensitive to the triploid block suppression suppressor by *admetos*”.

Reviewer #3:[…] The authors develop a specialized assay HIPOD-sco1 to detect polyspermy. This uses a mutation called sco1 that causes seed to be pale green. This is complemented by two different transgenes; (i) RPS5a::GAL4 and (ii) UAS::Tomato-SCO1. In the case of polyspermy both transgenes may be inherited causing complementation of sco1 seed phenotype and expression of the fluorescent TOM-SCO1 protein. I assume pollen is mixed from the two donors and then applied to the sco1 mutant flowers?

Thank you for this comment. No, we used the same technique applied in Nakel et al., 2017. This was not well specified in the current manuscript and we added the following text to the Materials and methods:"3 days after emasculation, pollination was carried out using pollen grains collected individually from plants of *sco1/- pRPS5a:mGAL4-VP16/+ and sco1/- pUAS::SCO1_tdTOMATO/+*. These two pollen categories were applied onto the stigmatic surface with two brushes."

The authors screened 56K seeds and found 10 seeds that showed evidence of complementation and tomato expression. Using spreads and labeling with CENH3-GFP the authors are able to confirm that the rescued embryos are triploid, and the endosperm is of expected triploid genotype (as in wild type), which implies that polyspermy of the egg cell and not the central cell occurred.Mutations in the ADM gene suppress the triploid block in Arabidopsis. The authors report the results of this experiment (subsection “Polyspermy-induced polyploidization events are often insensitive to triploid block suppression by admetos”) – is the data seen in adm1 significantly different from wild type. The comparison with data from interploidy crosses is also hard to interpret from the way this is presented.

Yes, we apologize for the confusion. The figure legend has been adjusted to ease understanding. We didn’t do any statistical analysis as we only recovered few data points. However, we now include in a table (Figure 3—source data 1) depicting the result gained in three independent experiments by three different scientists. From the data, it is evident that there is substantial consistency with respect to fold change.

In addition, we now more carefully phrase the *adm* results to avoid misunderstandings (see our response to reviewer #2).

Next the authors perform transcript profiling in triploid plants and consistent with published work observe few if any significant changes.Although the work is well performed – my major reservation is that I don't think a sufficient advance is made over previous work (Nakel et al., 2017) by the same authors.

Here we disagree. Until recently, hardly anybody expected triparental plants to inhabit our planet. In this study, we established a genetic detection assay that allows for the first time to determine the developmental origin of triparental plants. With this tool we were able to show selective polyploidization of the egg cell. This result is particularly unexpected as egg and central cell fertilization are coordinated during monospermy and because the egg cell block is stricter than the central cell block (Scott et al., 2008). Moreover, our finding that polyspermy can bypass the triploid block has important evolutionary and agricultural implications: First, it expands current polyploidization concepts, as it provides an explanation of how triploids can arise in the presence of a strict triploid block. Second, three-parent crosses might provide a means to overcome endosperm-induced hybrid incompatibilities by transmitting incompatible sperm along with compatible sperm through the egg cell only. We have rephrased two sentences in the Discussion in order to make this aspect clearer to the reader:

"Not only do three parent crosses hold the potential to instantly combine beneficial traits of three rather than two cultivars. […] Scrutinizing the potential of this genetic hitchhiking approach opens up new avenues for future research."

We would also like to refer to the comments made by reviewer 1 and 2 on the implication of our work.